# The Pleiotropic Role of Extracellular ATP in Myocardial Remodelling

**DOI:** 10.3390/molecules28052102

**Published:** 2023-02-23

**Authors:** Suhaini Sudi, Fiona Macniesia Thomas, Siti Kadzirah Daud, Dayang Maryama Ag Daud, Caroline Sunggip

**Affiliations:** 1Department of Biomedical Sciences, Faculty of Medicine and Health Sciences, University Malaysia Sabah, Kota Kinabalu 88400, Sabah, Malaysia; 2Health through Exercise and Active Living (HEAL) Research Unit, Faculty of Medicine and Health Sciences, Universiti Malaysia Sabah, Kota Kinabalu 88400, Sabah, Malaysia; 3Borneo Medical and Health Research Centre, Faculty of Medicine and Health Sciences, Universiti Malaysia Sabah, Kota Kinabalu 88400, Sabah, Malaysia

**Keywords:** extracellular ATP, purinergic receptor, purinergic signalling, myocardial remodelling, mechanism of action, drug repurposing, current pharmacological intervention in cardiovascular system

## Abstract

Myocardial remodelling is a molecular, cellular, and interstitial adaptation of the heart in response to altered environmental demands. The heart undergoes reversible physiological remodelling in response to changes in mechanical loading or irreversible pathological remodelling induced by neurohumoral factors and chronic stress, leading to heart failure. Adenosine triphosphate (ATP) is one of the potent mediators in cardiovascular signalling that act on the ligand-gated (P2X) and G-protein-coupled (P2Y) purinoceptors via the autocrine or paracrine manners. These activations mediate numerous intracellular communications by modulating the production of other messengers, including calcium, growth factors, cytokines, and nitric oxide. ATP is known to play a pleiotropic role in cardiovascular pathophysiology, making it a reliable biomarker for cardiac protection. This review outlines the sources of ATP released under physiological and pathological stress and its cell-specific mechanism of action. We further highlight a series of cardiovascular cell-to-cell communications of extracellular ATP signalling cascades in cardiac remodelling, which can be seen in hypertension, ischemia/reperfusion injury, fibrosis, hypertrophy, and atrophy. Finally, we summarize current pharmacological intervention using the ATP network as a target for cardiac protection. A better understanding of ATP communication in myocardial remodelling could be worthwhile for future drug development and repurposing and the management of cardiovascular diseases.

## 1. Introduction

Cardiac plasticity enables the heart to adapt in response to external or internal stimuli by changing in size. Myocardial remodelling occurs in response to various stimuli that can induce the heart’s morphological, metabolic, and molecular changes. This includes the cavity diameter and mass (to grow (hypertrophy) or to shrink (atrophy)), heart wall thickness, area of scar after injury, fibrosis, and inflammation. Adaptive remodelling, for instance, hypertrophic growth promoted during exercise, pregnancy, and postnatal growth, is due to the enlargement of myocardial size in response to an increase in protein synthesis and sarcomeres to compensate for increased demand [1]. However, these factors do not lead to heart failure because the increase in myocardial size is associated with increased contractility rather than a decrease in contractile function and energy metabolic production. In contrast, pathological hypertrophic remodelling occurs in response to hypertension, ischemia, myocardial injury, and fibrosis that can lead to cardiac dysfunction [2]. The hallmark of pathological myocardial remodelling is neurohormonal activation associated with progressive cardiomyocyte death, decreased energy metabolism and contractility, increased oxidative stress and inflammatory cytokines, fibrosis, hypertrophy, and vasoconstriction [3].

Extracellular nucleotides are messenger molecules that regulate numerous physiological and pathological processes in the cardiovascular system. Adenosine triphosphate (ATP) is an organic chemical that regulates numerous physiological mechanisms by acting on purinergic receptors [4,5]. In 1972, purinergic signalling was first proposed, where ATP plays roles as an extracellular signalling molecule and co-transmitter in sympathetic nerves [6]. Later in 1978, two distinct families of purinergic receptors, also known as purinoceptors, were identified based on their extracellular agonists [7]. Thereby, purinergic signalling was defined as the actions of extracellular purine nucleotides and nucleosides such as adenosine and ATP that mediate the activation of membrane-bound purinergic receptors. The roles of purinergic signalling have been widely reported in the regulation of cardiac function and the development of heart failure (Figure 1). 

Purinergic receptors are ubiquitously expressed in the cardiovascular system. ATP and other nucleotides, including ADP, UTP, and UDP, are released into the interstitial space in the event of the molecular adaptation of the heart to environmental changes, such as during mechanical stretch, chemical stress, and cell death [8]. Interestingly, the magnitude of ATP released from the stressed cells and its heterogeneities in the cardiovascular system may evoke a different physiological or pathological response depending on its sources, the purinergic receptors involved, and the downstream signalling. This review will discuss the mechanisms mediating ATP released from major cells constituting the cardiovascular system, followed by the paracrine signalling of ATP to the neighboring cells involved in pathological-remodelling-linked diseases. Finally, the therapeutic prospect of ATP to reduce or prevent pathological cardiac remodelling will be highlighted.

## 2. Extracellular ATP Sources and Signalling

The cardiovascular system is composed of cardiomyocytes and non-myocytes, including fibroblasts, vascular smooth muscle cells (VSMCs), endothelial cells (ECs), and circulating red blood cells (RBCs). Each of these cells communicates uniquely by releasing an autocrine/paracrine messenger known as ATP in response to different stimuli. Generally, in the cardiovascular system, ATP is released via the exocytotic mechanism and/or conductive channels. ATP-containing vesicles can be released by exocytosis through vesicular nucleotide transporter (VNUT) or lysosomal vesicles [9]. The exocytosis of these cytosolic vesicles is regulated by intracellular calcium (Ca^2+^) levels [10]. On the other hand, the conductive release of ATP is through ion channels, including the connexin hemichannels-formed gap junction, pannexin, volume-regulated anion channels, maxi-anion channels, ATP-binding cassette (ABC) transporter, and cystic fibrosis transmembrane conductance regulator (CFTR) [11]. The mechanism of ATP release from these channels depends on various effectors, such as the opening of connexin hemichannels, and is regulated by intracellular Ca^2+^, reactive oxygen species (ROS), and nitric oxide (NO) levels. In contrast, pannexin can be activated by mechanical stress and P2X7 receptor activation. 

Locally released ATP may act directly on the purinergic type 2 (P2) receptors expressed abundantly in fetal and adult human hearts to influence cardiac contractility [8]. Stimulation of ATP exerts inotropic and metabotropic effects via the stimulation of the P2X and P2Y receptors, respectively [12]. The ATP-sensitive P2X receptor family contains seven isoforms (P2X1-7) known as extracellular ATP-gated cation channels, whose activation regulates increased intracellular Ca^2+^ levels and contractility in rat hearts [13]. On the other hand, the P2Y receptor family contains eight isoforms (P2Y1, 2, 4, 6, 11–14) which are G-protein-coupled receptors (GPCRs) that form a specific isoform with either Gq, Gs, or Gi [14]. In the context of ATP-activated purinergic receptors on myocytes and non-myocytes in the cardiovascular system, some of these receptor-coupled G proteins share the same intracellular signalling cascade that is greatly involved in the physiological and pathological responses. In this section, we will highlight the source of ATP from myocytes and non-myocytes and its autocrine-mediated purinergic signalling in the heart. 

### 2.1. Cardiomyocytes

Extracellular ATP act as a co-transmitter in vagal cardiovascular reflexes. Mechanical stimulation promotes the opening of pannexin and connexin hemichannels to release ATP from the cardiomyocytes [15,16]. Gap junction channels composed of connexin provide a means of communication between adjacent cardiomyocytes for contraction coordination [17]. Among the connexin isoforms, the connexin-43 hemichannel is predominantly distributed in the heart, being the most prominent in the ventricles [18]. Under physiological conditions, the connexin-43 hemichannel localized in the sarcolemma of cardiomyocytes and remained closed. The gating of hemichannels is positively regulated by mitogen-activated protein kinase (MAPK) [19], protein kinase A (PKA) [20], and protein kinase C (PKC) [21], and protein phosphatase indirectly blunted the opening via dephosphorylation of the connexin-43 single channel [22]. Unlike connexin-43, the pannexin-1 channel does not involve gap junction formation but instead forms a protective functional pannexin-1/P2X7 complex, mediating the release of adenosine and ATP in a brief period of ischemic/reperfusion (I/R) [23]. Additionally, pannexin 2 is predominantly involved in ATP release in atrial myocytes in response to macrophage infiltration [24]. Several conditions and mediators promote pannexin-1 channel opening, including mechanical stress [25], ATP [26], cytoplasmic Ca^2+^ [27], and extracellular potassium [28]. In comparison, the channel closing is regulated by the negative feedback of ATP [29] and several channel blockers, namely carbenoxolone and probenecid [30]. 

In cardiomyocytes, the selectivity of ATP towards P2X receptors was determined by the effective concentration (EC_50_) value, which shows that P2X2, P2X4, P2X5, and P2X6 exert higher ATP selectivity than P2X7, but lower as compared to P2X1 and P2X2 [31,32]. Notably, P2X4 has been identified by several studies to be highly expressed in cardiac ventricular, hence its critical role in contractile performance [32,33]. A study by Musa’s group described the variation of cardiac P2X receptor distribution in different regions of different species [34]. The group recognized P2X4 and P2X7 as highly expressed receptors in the human right atrial and sinoatrial node using quantitative polymerase chain reaction and in situ hybridization. In contrast, a high abundance of P2X5 receptors was detected in the rat hearts’ left ventricle, right atrial, and sinoatrial node [34]. The physiological significance of ATP-mediated P2X receptor activation has been highlighted in an overexpression study of the P2X4 receptor in human ventricular myocytes. The stimulation of agonists ATP and 2-methylthioadenosine-5′-o-triphosphate (2-MeSATP) on the highly expressed P2X4 receptors augments basal cardiac contractility without provoking a hypertrophic-induced heart failure [32,35,36]. This beneficial cellular response was later supported by the pharmacological treatment of MRS2339, a hydrolysis-resistant adenosine monophosphate derivative that improves the cardiac output of in vitro and in vivo working heart models [37]. 

On the other hand, ATP is a natural selective agonist for four subtypes of P2Y receptors, including P2Y1, P2Y2, P2Y4, P2Y6, and P2Y11. These receptors are GPCRs that form a specific isoform with either Gq, Gs, and G12/13 alpha subunit [38,39]. The coupling of these receptors with Gq protein initiates the phospholipase C (PLC)/phosphatidylinositol-4, 5-bisphosphate (PIP2)/inositol triphosphate (IP3)-dependent Ca^2+^ mobilization and the activation of monomeric G proteins, Ras homolog family member A (RhoA), and Rac in cardiac muscle [40,41]. Additionally, ATP-induced P2Y11-Gs coupling yielded an increase in adenylate-cyclase-regulated cAMP production (Table 1). Interestingly, ATP-activated P2X6-G12/13 coupling in cardiomyocytes is revealed to be turned on right after the development of myocardial hypertrophy via the Gq-mediated pathway activated by the same receptor and agonist [42]. The G12/13 coupling mediates excessive production of fibrogenic factors, including transforming growth factor (TGF)-β, connective tissue growth factor (CTGF), and periostin in a Rho-dependent manner. This leads to the excessive deposition of extracellular matrix (ECM) proteins and, consequently, cardiac-fibrosis-induced heart failure.

### 2.2. Cardiac Fibroblasts

Cardiac fibroblasts (CFs) are a predominant non-myocyte cell type in the heart that orchestrates structural organization and corrects cardiac contraction by maintaining extracellular matrix (ECM) homeostasis. Interstitial fibrosis occurs at the earlier stage of cardiac remodelling in response to the changes in microenvironment dynamics by various stimuli [43]. At this stage, the deposition of ECM proteins, primarily collagens (type I and III), is reversible and necessary for sufficient scar formation associated with inflammatory reaction. CFs release several other pro-fibrotic activators to increase ECM synthesis, including angiotensin II (AngII), α-smooth muscle actin (α-SMA), cytokine transforming growth factor (TGF)-β, and plasminogen activator inhibitor (PAI)-1. 

An early study by Braun et al. (2010) characterized high activation of purinoceptor subtypes P2Y2 by ATP and UDP stimulation, increased collagen synthesis, and the expression of α-SMA, TGF-β, and PAI-1 in CFs [44]. The same group later revealed that connexin-43 and -45 hemichannel opening contributed to the release of ATP from CFs in response to hypotonic stimulation (Table 1). This substantially activates P2Y2-mediated pro-fibrotic marker production, including α-SMA, PAI-1, and monocyte chemotactic protein-1 (MCP-1), in an extracellular signal-regulated kinase (ERK)-dependent manner [45]. The group then discovered the counterbalancing effect of ATP release in adult rat CFs. The hydrolysis of extracellular ATP released endogenously from CFs produces adenosine, which provides an anti-fibrotic response dependent on adenosine subtype 2B (A2B) receptors and activation of the cyclic adenosine monophosphate (cAMP)-PKA pathway [46]. Overexpression of A2B receptors accompanied by an increased level of cAMP is critically involved in the inhibition of CF proliferation [47,48]. Additionally, another group reported that stimulation of A2B receptors yielded a significant reduction in endothelin-1 (ET-1)-induced a-SMA expression dependent on cAMP/Epac/PI3K/Akt signalling pathways in CFs [49]. Disturbance of collagen turnover implicates the balance between synthesis and degradation of ECM proteins, leading to irreversible replacement fibrosis, as can be seen in extensive ventricular remodelling associated with heart failure. The role of ATP in pathological-remodelling-induced cardiac fibrosis will be discussed later in this review.

### 2.3. Vascular Smooth Muscle Cells

Vascular smooth muscle cell (VSMC) proliferation is detrimental to vascular remodelling, growth, and the development of atherosclerotic cardiovascular disease [50]. The extracellular ATP acts as a potent stimulator for P2 receptors in the vascular smooth muscle cells (VSMCs), regulating blood vessel contraction and mitogenic effect. Pannexin-1 and connexin-43 are contributors to ATP release channels on VSMCs. Pannexin-1-released ATP regulates vasoconstriction intensity, triggered by the binding of phenylephrine on α1-adrenoreceptor [51,52]. ATP and other nucleotides are also released from the necrotic VSMCs due to the loss of membrane integrity [51,53]. In VSMCs, high ATP is released to act as a danger signal that alerts the circulating neutrophil to promote a wound-healing response. A study by Domenick’s group suggested that autocrine signalling of ATP contributed by the opening of connexin and pannexin hemichannels in response to mechanical stress in VSMCs. Treatment of non-selective hemichannel inhibitor carbenoxolone significantly attenuated the accumulation of ATP and its hydrolysis by-product inorganic pyrophosphate (PP_i_) [54]. Additionally, a class of integral membrane proteins known as ATP-binding cassette (ABC) transporters has been reported to play a functional role in releasing ATP [55,56]. Interestingly, these transporters are involved in regulating vascular tone in a cAMP/PKA-dependent vasorelaxation in VSMCs [57]. These findings provide a potential conduit for extracellular ATP accumulation targeting the VSMC purinergic receptors.

Uniquely, ATP possesses a dual phenotype-dependent effect on VSMCs [58,59]. ATP causes VSMCs to shift from a contractile into a proliferative phenotype in a concentration-dependent manner. A low concentration of ATP stimulates serum response factor (SRF), which enhances the expression of contractile specific proteins, smooth muscle 22 (SM22), and α-smooth muscle actin (αSMA). On the contrary, higher extracellular ATP concentration inhibits SRF activity and the specific contractile proteins, leading to the conversion of VSMCs into a synthetic phenotype [60]. Moreover, this phenotype shift is accompanied by the downregulation of the P2X1 receptor and the upregulation of mitogenic P2Y1, P2Y2, and P2Y6 receptors [61,62,63,64]. Continual release of ATP in response to high shear stress rapidly desensitized the P2X1 receptor, blunted its expression, and substantially promoted vascular wall relaxation prior to vascular remodelling via activation of mitogenic Gq-protein-coupled P2Y receptors [65,66] (Table 1). Hogarth et al. later proposed the downstream mechanism where ATP-induced P2Y receptors provoke a transient activation of the cAMP-PKA pathway, yielding an inhibitory effect on SRF activity [67]. 

### 2.4. Vascular Endothelial Cells

The inner cellular lining of arteries, veins, and capillaries comprises a monolayer of endothelial cells (ECs). The vascular endothelium directly interacts with the circulating blood and is considered a barrier between the blood and the vessel wall. In such an arrangement, ECs are exposed to the mechanical force generated by flowing blood (shear stress), which can exert significant autocrine, paracrine, and endocrine actions influencing VSMCs, platelets, peripheral leucocytes, and circulating RBCs [68]. The dynamic role of ECs in the vascular system is depicted by the involvement in the synthesis of vasodilator factors (NO and prostacyclin) [69,70], vasoconstricting factors (angiotensin-converting enzyme (ACE), endothelin (ET), and free radicals) [71], inflammatory mediators (interleukins (ILs) and major histocompatibility complex (MHC)) [72], and growth factors (insulin-like growth factor (IGF) and transforming growth factor (TGF)) [73]. In addition to these mediators, EC-released ATP under shear stress is mediated by the influx of extracellular Ca^2+^ via P2X4 receptors [74,75]. 

Purinergic receptors are widely distributed in the vascular system, and endothelial cells express multiple receptor subtypes, indicating their physiological and pathophysiological importance. There are several mechanisms involving ATP release channels in ECs, including caveolin-1 (Cav-1) [76], volume-regulated anion channels (VRAC) [77], and connexin hemichannels [78]. Interestingly, shear stress evoked an increase in intracellular Ca^2+^ where the release of ATP is localized. Yamamoto’s group revealed that the ATP release from ECs always preceded the Ca^2+^ increase [76]. This is supported by the action of ATP-activated purinergic receptors, which contribute to elevated intercellular Ca^2+^ levels via P2X4 and P2Y1, 2, 4, 6-coupled Gq-PLC pathways [74,79,80]. In this cellular context, the cytosolic pool of Ca^2+^ under shear stress leads to endothelial nitric oxide synthase (eNOS)-NO-mediated vasodilation (Table 1). Although NO production depends on Ca^2+^ level, an additional mechanism of eNOS phosphorylation by PKA, protein kinase B (PKB), and cyclic guanosine monophosphate (cGMP)-dependent protein kinase II (cGKII) may compensate for NO production in the event of intercellular Ca^2+^ reduction [81,82]. Disruptions of the regulation of these channels contributed to vasoconstriction.

### 2.5. Circulating Red Blood Cells

The release of cytosolic ATP from circulating red blood cells (RBCs) is reflected in the microenvironment, such as under the hypoxic condition [83], mechanical deformation [84], shear stress [85], and upon immune adherence clearance mediated by ligation of complement receptor 1 (CR1) on RBCs [86]. Throughout the decades, the mechanism of ATP release in RBCs has been revealed via several channels. A membrane-bound nucleoside transporter of RBCs is the earliest channel reported to export ATP under hypercapnic conditions [87]. Subsequent studies by Sprague’s group revealed the involvement of cystic fibrosis transmembrane conductance regulator (CFTR)-dependent adenylyl cyclase-cAMP pathway [88,89]. The same group further demonstrated that activation of CFTR and pannexin-1 hemichannel contributed to ATP release [90,91], which was supported by other works following RBCs’ deformation [92,93]. Independently, activation of the cAMP-PKA pathway promotes oligomerization of voltage-dependent anion channel (VDAC) with the translocase protein TSPO2 and nucleotide transporter that transiently accumulate extracellular ATP up to 1-2 µM within a few minutes [94]. 

Like other cells in the cardiovascular system, purinergic receptors, particularly P2 receptors involved in ATP-mediated signalling, are abundantly distributed in the RBCs [95]. Quantitative PCR of human RBCs by Wang et al. revealed the expression level of P2 receptors in ascending order as follows: P2Y1/P2Y4/P2Y6/P2Y11/P2Y12 < P2X1/P2X4, P2X7/P2Y2 < P2Y13 [96]. From a teleological perspective, the non-physiological release of ATP may contribute to adaptive or maladaptive responses. For instance, bacterial toxin mediates autocrine signalling of ATP to act on P2X1 and P2X7, promoting phosphatidylserine (PS) exposure on RBCs that leads to cell shrinkage [97]. Furthermore, activation of the complement system in the innate immune response exacerbates the ATP-P2X-mediated hemolysis, for which the downstream mechanism remains to be elucidated [98]. Regardless, the ramification of ATP roles on P2X activation poses a high risk of blood-related diseases, including anemia, leukemia, or autoimmunity. On the other hand, the functional role of ATP on purinergic signalling has been revealed by several studies on P2Y receptors. Several studies demonstrated the involvement of P2Y1 receptors in malarial parasite development. Autocrine activation of P2Y1 by ATP induces an osmolyte permeability pathway in human and murine RBCs, providing sufficient nutrients for the erythrocytic cycle of the parasites [99,100] (Table 1). 

Interestingly, higher expression of P2Y13 in human RBCs was revealed to negatively regulate ATP release from these cells. Wang and group depicted that the metabolism of ATP to ADP, which act on P2Y13 receptors, impairs the release of ATP via the inhibition of cAMP [96]. As mentioned earlier, this inhibition may blunt the opening of CFTR and VDAC, which are cAMP-dependent ATP release channels. Moreover, paracrine signalling of ATP release from RBCs stimulates the P2 receptor on the vascular endothelium, which can promote endothelium-dependent and smooth-muscle-dependent vasodilation via the generation of vasodilators and anti-inflammatory factors such as nitric oxide (NO) and prostacyclin (PGI2) [101]. Additionally, NO also inhibits hypoxia-induced ATP release from RBCs [102].

## 3. ATP in Cardiovascular Remodelling

ATP is part of the damage-associated molecular patterns (DAMPs) metabolites released during cellular stress and from dead or dying cells. ATP acts as a major signalling molecule in purinergic signalling, where it can directly activate P2X and P2Y receptors. Moreover, rapid degradation of extracellular ATP to ADP, AMP, and adenosine stimulates the activation of various P1 and P2Y receptors, which initiates intracellular signalling pathways in the pathological remodelling of the heart. Cell-to-cell communication becomes a crucial checkpoint of remodelling in the highly complex cardiovascular system. Cell communication is divided into paracrine/autocrine signalling, direct cell-to-cell interaction, and extracellular matrix interactions. ATP may act as a communication signal between resident cells in the cardiovascular system, for instance, VSMCs-ECs partners in hypertension and atherosclerosis, RBC-ECs-VSMCs-cardiomyocytes–cardiac fibroblast in ischemic/reperfusion, and fibroblast–cardiomyocyte crosstalk in myocardial fibrosis and cardiac hypertrophy. Numerous pieces of evidence that connect direct and indirect contact of these cells via ATP signalling will be further discussed in the context of pathological cardiovascular remodelling.

### 3.1. Hypertension and Atherosclerosis

Hypertension is a major risk factor for developing congestive heart failure, promoting myocardial inflammation, hypertrophic vascular remodelling, and atherosclerosis. In the wake of the COVID-19 pandemic, hypertension contributes to the complication and mortality of this viral infection. The multi-interaction of several systems, including the sympathetic nervous system, the renin–angiotensin–aldosterone system, the endothelium, and the immune system in hypertension, demonstrated the intricate signalling networks in this vascular pathology. The development of hypertension begins at the abnormalities of vascular volume regulation that further enhance the vasoconstriction and subsequently cause arterial remodelling by decreasing lumen diameter and increasing resistance [103]. Low-grade chronic inflammation is a vital point in hypertension. Of importance, endothelial dysfunction and hypertrophic VSMC remodelling critically emanate vascular complications via impairment of vascular tone regulation, oxidative stress, and inflammation due to the close proximity of these two vascular residents [104,105]. 

Accumulating evidence highlighted the importance of P2X7 activation, specifically stimulation by ATP, as a significant modulator in the sterile inflammatory response of hypertension. In the event of hypertension, local accumulation of ATP surges from the vesicle in peripheral sympathetic nerves [106], ECs [76], VSCMs [52], and RBCs [85]. Shen et al. reported that after the onset of hypertension, extracellular ATP reached up to 3 µM in mice [107]. Clinical data also supported a substantial increase in plasma ATP in untreated hypertensive patients compared to treated and normotensive control patients [107,108]. In other studies, ATP exacerbates myocardial inflammation during hypertension via stimulation of P2X7 receptors in VSCMs and ECs. This activation mediates nucleotide-binding domain (NOD)-like protein 3 (NLRP3) inflammasome to release pro-inflammatory cytokines, including interleukin-1 beta (IL-1β) and interleukin-18 (IL-18) [109,110,111]. Furthermore, in cardiomyocytes, ATP-mediated P2X7 activation contributes to pyroptosis by the action of caspase-1-mediated pore-forming protein Gasdermin-D cleavage, which also facilitates the release of IL-1β and ATP from the dying myocytes [112]. Additionally, a long-term increase in ATP acts as DAMP metabolites to recruit immune cells in autoimmune features. This action directly evokes antigen-presenting cells (APCs)-mediated overactivity of T cells, elevating the blood pressure, which is the earliest mark of hypertension [108]. It is worth noting that P2X7 exerts a regulatory effect on blood pressure, which was shown with the improvement of systolic and diastolic pressure of hypertensive P2X7 receptor knockout mice. Therefore, these findings indicate that P2X7 is a promising target for the treatment of hypertension. 

In a spontaneously hypertensive rat model, elevated sympathetic-driven vasoconstriction is dependent on ATP activation of purinergic receptors [113]. This is supported by the clinical data showing the association of hypertension with enhanced sympathetic nerve activation, promoting vasoconstriction due to hypertrophic VSMC remodelling [114,115]. In regulating vascular remodelling during hypertension, ATP signalling on the P2 receptor subtypes is reported to be tissue/cell- and period-dependent. For instance, the immediate reaction of ATP would act as a neurotransmitter to activate P2X4 and P2YRs (P2Y2 receptor as a major subtype other than P2Y1) on ECs. This contributes to the production of vasodilators such as NO and prostacyclin, leading to vasodilation and a decrease in blood pressure [79,80,116]. Notably, previous studies demonstrated that knockout of the P2X4 receptor induces hypertension in mice by attenuating ATP-induced Ca^2+^ influx, impairing endothelial production of NO and vasodilation [75,117]. Although ATP action on endothelial P2Y2 receptors is physiologically important in promoting vasodilation and relaxation [118,119], a study by Chen’s group emphasizes the pathophysiological importance of long-term activation of this receptor in atherosclerosis [116]. EC-specific deletion of P2Y2 reduces p38- and Rho kinase-dependent expression of vascular cell adhesion molecule-1 (VCAM-1) and increases VSCM migration through the inhibition of eNOS production [116,120]. These findings depicted the pro-inflammatory role of ATP-induced endothelial P2Y2 receptor in plaque formation and macrophage infiltration in the manifestation of atherosclerosis. 

On the other hand, long-term accumulation of ATP promotes P2X1, P2Y2, and P2Y6 activation on VSMCs, which mediates vasoconstriction via ROS or thromboxane A2 (TxA2) production [121,122]. Stimulation of P2X1 mediates Ca^2+^-influx-dependent vasoconstriction [123], while P2Y2 and P2Y6 promote vasoconstriction via the release of intracellular Ca^2+^ storage [124]. The tightly regulated minute-to-minute vasoconstriction event is potentiated during thrombocyte aggregation at sites of severe atherosclerosis in response to ATP action on P2X receptors located on VSCMs. This activation is unchallenged by P2-mediated vasodilation on ECs due to endothelial damage [125]. A subsequent study by Harhun’s group identified that sustained ATP stimulation evokes vasoconstriction by rapidly desensitizing homomeric P2X1 receptors and heteromeric P2X1/P2X4 receptors in rat VSMCs [126]. Additionally, ATP acts as a growth factor on P2Y2 and P2Y6 receptors mediating VSMC proliferation and growth, a key event in the development of hypertrophic vascular remodelling, hypertension, and atherosclerosis [65,127]. Moreover, increased P2Y6 abundance forms a functional coupling with angiotensin AT1 receptors contributing to age-dependent hypertension in adult VSMC mice, as reported by Nishimura and colleagues [128]. The group suggested the underlying mechanism via G-protein-dependent ERK and Akt activation to promote hypertrophic growth. Similar to ECs, ATP action on VSMC P2Y2 receptors also facilitates early migration of VSMCs to the intima by releasing metalloproteinase-2, which induces matrix degradation [129]. Furthermore, activation of VSMC P2Y2 receptors promotes its proliferation via ERK1/2- and PI3K-dependent mechanisms during hyperplasia associated with hypertension [130]. 

Apart from potently modulating P2 receptors on the cardiovascular resident cells during hypertension, the compensatory action of ATP stimulating other purinergic receptors is also reported in the role of cardiovascular protection. For instance, ECs released massive amounts of ATP in response to high blood pressure, activating P2X4 on neighbouring ECs to rapidly generate NO, a potent vasodilator [14]. Furthermore, several studies have reported the activation of adenosine receptors, which modulates the adaptive response to decrease blood pressure and hypertension. Activating the A2A receptor mediates vasodilation in ECs and VSMCs by increasing production of NO and cAMP, respectively, increasing blood flow [131,132] (Table 2). In this cellular context, increasing the rate of ATP metabolism to adenosine may seem to be a favorable therapeutic strategy. However, further understanding of the coordination and communication between RBCs, ECs, and VSCMs via purinergic signalling of ATP is critical in understanding the complex mechanism during hypertension.

### 3.2. Ischemic Reperfusion (I/R) Injury

Acute myocardial infarction occurs in the event of coronary artery obstruction. Reduction in blood flow or ischemia blunted the nutrient and oxygen transportation to the heart, leading to hypoxic conditions and, consequently, cardiomyocyte death. Re-establishing the blood flow may be preferable to avoid further cellular damage. Paradoxically, this method resulted in massive drawback, as a sudden increase in oxygen supply mediates the release of reactive oxygen species (ROS), ultimately leading to cardiac damage known as reperfusion. Extracellular ATP is detrimental to promoting tissue inflammation during I/R. Under ischemic and hypoxic conditions, ATP is released from swelling cells via cardiac maxi-anion channels [133], apoptotic cardiomyocytes, ECs, and circulating RBCs via pannexin-1 hemichannels [134,135,136], and activated neutrophil and EC via connexin-43 hemichannels [137]. In addition, ischemia induces the opening of pannexin-1 and the release of ATP from cardiac fibroblasts, causing them to transform into myofibroblasts [138]. Accumulating extracellular ATP pathologically activates P2X7-dependent NLRP3 inflammasome in macrophage and fibroblast [138,139] and P2Y6-induced vascular inflammation in EC [140,141]. Although it was suggested that blocking ATP release might be a therapeutic strategy to prevent tissue injury, several studies have reported that the protective role of ATP may compensate for these deleterious effects during I/R.

In cardiac muscle, P2Y2 appears to be highly expressed in the left ventricular of congestive heart failure [142]. Remarkably, activation of P2Y2 receptors by both ATP and uridine triphosphate (UTP) exerts a protective role during I/R [143,144]. Like ATP, UTP is released during cardiac ischemia, and 48 h pre-administration of UTP is reported to reduce infract size and improve mice heart function mediated by P2Y2 and P2Y4 receptor activation [145]. Although the downstream mechanism of ATP/UTP-mediated P2Y2 activation in protection against I/R is not fully elucidated, it most likely involves the transactivation of several survival kinases, including PI3K, Akt, and ERK [146]. These kinases critically participate in the blunted formation of mitochondrial permeability transition pore that causes ATP depletion and formation of ROS, promoting organelle rupture and further myocyte necrosis. Additionally, ATP exerts a positive inotropic effect through the formation of a non-selective pannexin-1/P2X7 channel complex which contributed majorly to the release of endogenous cardioprotectants such as sphingosine 1-phosphate (S1P) and adenosine [23,147]. Moreover, a recent study by Matsuura’s group proposed that the autocrine release of ATP via the maxi-anion channel contributes to the recovery of left ventricular function [148]. 

Interestingly, the rapid breakdown of accumulated extracellular ATP to adenosine enhances protection against myocardial ischemia. It has been suggested that the downregulation of tissue-specific and systemic inflammatory response is via a negative feedback mechanism of A2 adenosine receptors [149]. In a murine model of ischemic pre- and post-conditioning, the A2B adenosine receptor in cardiomyocytes and ECs is activated by adenosine. This subsequently prompted the attenuation of infarct size and vascular leakage [150,151,152] (Table 2). A robust and selective activity of ectonucleoside triphosphate diphosphohydrolases (E-NTPDases), also known as CD39, during hypoxia escalates the conversion of ATP to AMP [153]. Later, ecto-5′-nucleotidase CD73 catalyzed the conversion of AMP to adenosine, which was also transcriptionally induced during myocardial ischemia by the regulation of hypoxia-inducible factor 1α (HIF1α) [152]. Stable expression of HIF1α protects against I/R by mediating the adaptive response of the cells to the hypoxic condition. Remarkably, the actions of ATP and adenosine on P2Y2 and A2B receptors were also found in ECs, which significantly protect against hypoxia-induced endothelial apoptosis and dysfunction [154].

Although ATP may seem to protect against I/R injury, accumulation of nucleotides (especially ATP) in prolonged hypoxia may reach a toxic level and consequently promote apoptosis in cardiomyocytes and excessive fibrotic response [155]. In time, this will likely lead to life-threatening changes in cardiac muscle wall structure and contractility. Further in vivo studies are needed to determine how the positive and negative effects of P2 receptor stimulation by ATP and adenosine balance out during myocardial I/R injury.

### 3.3. Angiogenesis

Angiogenesis is one of the therapeutic strategies to alleviate cardiac damage following ischemic events. This process is vital to help in wound healing and delivering blood, oxygen, and essential nutrients to the injury site following a series of the ischemic event [156]. The communication between cardiomyocytes and ECs residing in the microvasculature is critical to fulfill the oxygen and metabolic demand from both physiological and pathological stimuli. Therefore, the coordination of tissue growth and angiogenesis in the heart is vital to prevent the progression of heart failure. Coronary angiogenesis is an important determinant for the transition from adaptive to pathological cardiac hypertrophy [157]. In the acute phase of adaptive cardiac growth, Akt modulates angiogenesis via enhanced expression of potent myocardial angiogenic factors, including vascular endothelial growth factor (VEGF) and angiopoietin-2, in an mTOR-dependent manner. Later, the inhibition of angiogenesis by prolonged activation of Akt downregulates the mTOR-dependent VEGF and angiopoietin-2 expression, leading to contractile dysfunction and pathological hypertrophy [158]. On the other hand, in ECs, short-term activation of Akt-coordinated VEGF expression attenuates the damage by ischemia. In contrast, prolonged activation of Akt promotes unorganized blood vessel formation, as can be seen in tumor vasculature [159]. Therefore, induction of angiogenesis via VEGF-VEGF receptor 2 signalling has been recognized as an attractive strategy in myocardial infarction, whereby the salvation of dying cardiomyocytes takes place. A dynamic communication between cardiomyocytes and ECs occurs via connexin 43 [160]. This suggests that a VEGF-dependent mechanism induces the formation of vasculature and stimulation of VCAM-1, contributing to endothelial survival, proliferation, and migration [161]. 

A study by Seye’s group reported that P2Y2 receptor promotes VCAM-1 expression via direct interaction with the VEGF receptor, which mediates activation of the Rho exchange factor in human coronary artery ECs [162]. This early finding indicates a direct link between purinergic and angiogenesis signalling. Dynamic interaction between ATP and VEGF was later revealed to contribute to the stimulation of an angiogenic response in a temporal- and environment-dependent manner [163]. Furthermore, during chronic and transient ischemia, elevated VEGF expression potentiates the VEGF-mediated intracellular Ca^2+^ increase via TRPC3 and TRPC6 channels on ECs [164,165]. Evidently, the pro-angiogenic role of extracellular ATP is revealed to act synergistically with VEGF via P2Y2 receptors with a critical downstream activation of ERK1/2, PI3K/Akt, and mTOR signalling pathways in the hypoxic microenvironment of ECs [166]. In line with this finding, a recent study by Muhleder’s group revealed the shear-stress-induced ATP release from ECs, which in turn act on P2Y2 receptor in an autocrine manner, leading to not only VEGF-induced angiogenesis but also NO-induced vasodilation [167]. Interestingly, the studies that were conducted on human umbilical vein ECs and induced pluripotent stem cell-derived ECs showed a similar finding on the functional role of P2Y2 activation in angiogenesis. Additionally, a finding by Gast et al. [168] shows that extracellular ATP forms a functional complex with VEGF isoform 165, contributing to the proliferation of ECs. This finding further emphasizes the significant role of ATP in angiogenesis, which can act independently from purinergic signalling. 

Although promoting angiogenesis is favorable in preserving myocardial contractile function and improving recovery during the post-ischemic event, a compelling study has demonstrated its detrimental role in tumor progression. The pro-metastasis role of extracellular ATP in angiogenesis has been highlighted in triple-negative breast cancer (TNBC). The ATP-P2Y2 signalling axis promotes angiogenesis and TNBC cell adhesion to ECs via the co-activation of CTGF, VEGF, and VCAM-1, thereby facilitating TNBC progression and metastasis [169]. It was suggested that elevated extracellular ATP activates P2Y receptors above the threshold, contributing to the VEGF-VEGF receptor 2 angiogenic signalling axis, promoting pathological angiogenesis in tumor progression [170]. Furthermore, ATP release from ECs may activate the P2X7 receptor on circulating monocytes that trigger the production of VEGF-induced tumor angiogenesis [171]. Contrary to these findings, a study by Gidlof’s group [172] revealed an opposite role of the ATP-P2Y2 axis in ECs, which activates the microRNA-22 (miR-22) promoter, leading to the downregulation of ICAM-1, a major endothelial adhesion molecule responsible for vascular inflammation (Table 2). Recent findings later demonstrated the role of miR-22 as a potent hepatic tumor suppressor [173], suggesting the ATP-P2Y2-miR-22 signalling cascade might be beneficial in combating tumorigenesis. Given the importance of angiogenesis, therapeutic targeting of the purinergic signalling in a patient with tumor progression and a heart condition will be challenging. Therefore, targeting the cell-specific biomarker that produces pro-angiogenic factors will be necessary for designing suitable microcirculation therapies.

### 3.4. Myocardial Fibrosis

Due to the negligible regenerative capacity of the adult cardiomyocytes, reparative fibrosis takes place to preserve the structural integrity of the infracted ventricle in the early event of cardiac injury. Cardiomyocytes respond to the increase mechanical load through a process known as mechanotransduction, and its downstream effect functions as a compensatory adaptive response by initiating the reactive interstitial fibrosis to preserve the cardiac structure. However, fibrosis becomes a pathogenic response in an extensive myocardial remodelling that disrupts cardiac morphology, ECM deposition, and impaired systolic and diastolic function [174]. In the absence of infarctions, prolonged and abnormal loading conditions may act as pathophysiological stimuli that change cardiac myocyte growth regulation via hypertrophic or atrophic response. In the event of cardiac injury, activated fibroblast transdifferentiates into myofibroblasts, and the process known as replacement fibrosis occurs to make up for the death of cardiomyocytes. As the cardiac disease progresses, the myofibroblast population predominantly contributes to the resident cardiomyocytes’ structural, biochemical, mechanical, and electrical properties. Recent findings revealed that the myofibroblast–myocyte coupling compromised mechanical activity of cardiomyocytes by reducing their contractility due to reducing cell length shortening [175,176]. These findings revealed the underlying consequences of how the increased myofibroblast/fibrosis can cause dysfunctional heart contractility in myocardial infarction.

At the injury site, resident myofibroblasts produce excessive ECM proteins, collagen, and pro-inflammatory cytokines [177]. Additionally, myocytes and non-myocyte cells release nucleotides and other pro-inflammatory factors that trigger the inflammatory responses. Since fibroblasts possess an ability to sense the danger signals released in the extracellular environment, the communication signalling of fibroblasts with myocytes and non-myocyte cells is vital to begin the fibrotic process at the site of an injury. For instance, following myocardial infarction, robust ATP released via pannexin-1 channels from cardiomyocytes, RBCs, and ECs acts as a paracrine signal to initiate inflammation-mediated fibroblast activation [90,136]. Dolmatova’s work suggested that ATP mediates fibroblast–myofibroblast transformation via upregulation of pro-fibrotic MAPK and p53 pathways [138]. High ATP release also contributes to IL-1β accumulation, a potent activator of p38 MAPK/Akt-regulated fibroblast proliferation and migration [138]. Noteworthy is the fact that autocrine signalling of ATP release from connexin-43 hemichannels of the resident fibroblasts also significantly contributed to the pro-fibrotic response via the activation of P2Y2-ERK signalling axis-mediated α-SMA and collagen accumulation [45]. Consequently, the activation of these pathways leads to increases in α-SMA, collagen type III, and TGFβ, aggravating the fibrotic response [44]. 

Apart from the pro-fibrotic activation, the ATP-P2Y2 axis has been reported by several studies to modulate Ca^2+^-induced ROS production, which causes replacement fibrosis upon cardiomyocyte death. Diacylglycerol (DAG) is the other second messenger that PLC produces in ATP-P2Y2-Gq signalling [178]. DAG directly activates TRPC3-mediated Ca^2+^ influx and functions as a positive regulator of reactive oxygen species (ROS), leading to mechanical-stress-induced maladaptive fibrosis [179,180,181]. TRPC3 predominantly influences pressure overload cardiac remodelling and cardiac fibrosis instead of cardiac hypertrophy by activating Nox2-derived ROS signalling in cardiomyocytes and fibroblasts [180]. TRPC3 interacts with Nox2, a membrane-bound subunit activated by cytoplasmic subunits p47^phox^, p40^phox^, p67^phox^, and small GTPase Rac1 and Rac2 [182]. Nox2 is reported to participate in cardiac remodelling after myocardial infarction and age-associated cardiac remodelling [183,184]. A negative regulator of ROS (NRROS) localized in the endoplasmic reticulum competes with p22^phox^ to bind to the gp91^phox^ subunit and facilitates the degradation of Nox2 through endoplasmic-reticulum-associated degradation [185]. However, the direct interaction of TRPC3 with p22^phox^ stabilized Nox2 and protected the enzyme from proteasome-dependent degradation. Mechanical stretch upregulates TRPC3, which activates PKCβ to recruit organizer subunit p47^phox^ for Nox2-dependent ROS production amplification, leading to the onset of left ventricular dysfunction [180]. Furthermore, in a different pathological setting, the interaction of the TRPC3-Nox2 complex is associated with the cardiotoxicity effect of doxorubicin, which forces the heart to shrink due to myocardial apoptosis and interstitial fibrosis at a later stage of left ventricular dilated cardiomyopathy [186].

Another purinergic receptor that is highly expressed in the activated cardiac fibroblast is P2X7. The inotropic action of ATP on P2X7 receptors contributes to the increase in cytosolic Ca^2+^ directly via P2X7-mediated Ca^2+^ influx and indirectly via Ca^2+^-activated Ca^2+^ channels through store-operated ion and TRP channels [187]. Interestingly, the Ca^2+^ signalling mechanism is a prerequisite in the fibroblast-to-myofibroblast differentiation via the TRPC6-Ca^2+^-calcineurin-NFAT signalling cascade [188,189]. Moreover, TRPC3-mediated Ca^2+^ signalling via activation of the ERK 1/2 signalling cascade also contributed to the proliferation and differentiation of fibroblast [190]. In addition to mediating fibroblast differentiation, P2X7 also aggravates the inflammatory response in the pathological remodelling of cardiac fibrosis. ATP-stimulated P2X7 receptors in cardiac fibroblast induce cellular activation of the NLRP3 inflammasome, increasing IL-1β production [138]. Evidently, pharmacological inhibition and silencing of P2X7 significantly suppressed abnormal cardiac fibroblast activation in mouse models of pressure-overload-induced myocardial remodelling [191].

On the other hand, activation of the P2Y1 receptor is reported to show a protective role in the development of myocardial fibrosis. A recent finding by Tian et al. demonstrated the downregulation of P2Y1 expression in TAC-induced cardiac hypertrophy and fibrosis [192]. Consistently, P2Y1 expression also decreased in TGFβ-activated cardiac fibroblast, and silencing the P2Y1 gene expression further escalated the activation of myofibroblast and the process of fibrosis [192] (Table 2). An early study by Franke’s group supported this finding, where the ATP-P2Y1 axis protects against oxidative-stress-induced cell death by modulating the pro-fibrotic ERK1/2 phosphorylation to the basal level [193]. Collectively, suppressing the over-activation of MAPK signalling pathways (ERK and p38) potentiates the impaired production of α-SMA, CTGF, and ECM depositions.

### 3.5. Hypertrophic and Atrophic Remodelling

A variety of stimuli and environmental stressors instigate cardiac remodelling by either inducing the heart to grow (hypertrophy) or shrink (atrophy). Physiological cardiac growth, as seen in endurance and resistance training, increases terminally differentiated cardiomyocyte mass without causing any increase in cell number [194]. On the other hand, cardiac atrophy is usually a complication of other events such as prolonged bed rest, weightlessness during space travel, and pharmacological or surgical intervention to stimulate ventricular unloading, such as the use of a left ventricular assist device (LVAD) [195]. Cancer-induced cardiac atrophy is a ramification that can result from cancer itself and various cancer chemotherapies. Rodent models of cancer-induced cardiac atrophy showed characteristics including a high rate of protein degradation that, in turn, causes a decrease in cardiomyocyte size, which is responsible for a total reduction in myocardial mass [196]. Although different adaptation responses trigger cardiac hypertrophic and atrophic remodelling, both share many common cellular and molecular signalling profiles.

External factors such as insulin and insulin-like growth factor (IGF1) activate the PI3K/Akt pathway, which is the critical center in cardiac molecular adaptations to exercise. PI3K was identified as a critical mediator in exercise-induced hypertrophy that exhibits a cardioprotective role to attenuate pressure-overload-induced pathological hypertrophy [197]. On the other hand, the development of myocardial hypertrophy comprises complex cellular and molecular events within cardiomyocytes and non-myocytes, including vascular ECs, VSMCs, fibroblasts, and immune cells that release neurohumoral factors such as ET-1, TGF-β, AngII, cytokines, and nucleotides. These factors act as key pathogenic determinants of myocardial hypertrophy via the stimulation of G-protein-coupled receptors, essentially the Gq family in a PLC/Ca^2+^-modulated hypertrophic gene expression in cardiomyocytes [198,199]. 

Based on previous studies, ATP-induced high cytosolic Ca^2+^ binds to calmodulin and further activates calcineurin, leading to increased activity of nuclear factor of activated T (NFAT) through dephosphorylation [200,201,202]. Although ATP action on purinergic receptors induces classic hypertrophic signalling pathways, such as an increase in intracellular Ca^2+^ and MAPK via Gq-coupling-activated PLC signalling pathway in cardiac cells, ATP failed to promote cardiomyocyte hypertrophy [201,202]. The underlying mechanism was later revealed by Sunggip et al., where ATP induces a sustained increase in intracellular Ca^2+^ level but not a transient increase, as can be seen with the treatment of hypertrophic agents such as AngII and ET-1, hence the failure to produce a hypertrophic response. The group also revealed that ATP negatively regulates hypertrophic response in cardiomyocytes by the functional coupling of transient receptor potential canonical 5 channel (TRPC5) and endothelial nitric oxide synthase (eNOS) [200]. TRPC5-eNOS coupling blunted the TRPC3/TRPC6-mediated Ca^2+^ influx via the NO-cGMP-PKG axis. This contributes to sustaining the increase in intracellular Ca^2+^, which is not sufficient to activate the nuclear factor of activated T cell (NFAT)-dependent hypertrophic response.

Interestingly, P2X4, another ATP-responsive receptor in cardiomyocytes, is found to be beneficial in increasing the cardiac contractile function via the interaction with eNOS in mouse models of pressure overload heart failure [203]. Potentially, this interaction may contribute to the inhibition of TRPC3/TRPC6 channels. Additionally, previous studies have consistently shown the essential role of GPCR-stimulated Ca^2+^ signalling through DAG-activated TRPC3 and TRPC6 in myocardial hypertrophy and interstitial fibrosis [180,181,204]. These findings demonstrated the potential counter-regulatory effect of ATP inhibiting the progression of cardiac hypertrophy.

A finding from our laboratory later revealed that the activation of the P2Y2 receptor by ATP exhibits an atrophic response in neonatal rat cardiomyocytes, indicated by the upregulation of muscle atrophy F box (MAFbx or atrogin-1) protein [205]. We also reported that a high concentration of ATP was released in response to 6 h of exposure to pathophysiological stresses involving hypoxia, glucose, and amino acid starvation, inducing cardiomyocyte atrophy via TRPC3 and Nox2 complex formation [205] (Table 2). Fasting- and starvation-induced nutrient deprivation have been linked to activating the ubiquitin-proteasome pathway (UPP) and other atrophic programs such as autophagy and apoptosis [206]. MAFbx was identified to play an essential role in mediating atrophy and suppressing hypertrophy in cardiac muscle. The negative regulation of MAFbx on the hypertrophic response is via the inhibition of MAPKs (ERK1/2, JNK1/2, and p38) and NF-κB signalling pathways [207]. Collectively, we postulate that high ATP production/treatment may initiate the atrophic response as an initial compensatory mechanism, thereby counteracting the hypertrophic signalling in cardiomyocytes. However, prolonged accumulation of ATP in the extracellular matrix may potentiate the protein turnover in favor of the degradation pathway, leading to cardiomyocyte death, increased myocardial interstitial fibrosis, and heart failure. 

## 4. Therapeutic Insights of ATP Signalling in Cardiac Remodelling

The pleiotropic role of ATP modulating protection and harm mediated by P2 receptors in several cardiovascular diseases undeniably adds to the purinergic signalling complexity. Early in the 1940s, angina-pectoris-associated coronary disease was treated with ATP injections, and additionally, ATP was also used to treat patients with coronary insufficiency [14]. Treatment with a low concentration of ATP (30 µM) has been proven to inhibit ischemic pre-conditioning [147]. These findings further emphasize the therapeutic potential of this nucleotide. On the contrary, several findings also highlighted the contribution of extracellular ATP released by stressed or dying cardiomyocytes and other resident non-myocytes in the pathological progression of cardiac diseases. Due to the complex cell-to-cell communication in cardiovascular signalling, targeting one specific modulator may nullify other downstream pathways that might be critical for the protective mechanism. For instance, blocking ATP action in an excessive inflammatory response associated with myocardial ischemia might blunt the protective roles of ATP in modulating downstream survival pathways and adenosine-modulated anti-inflammatory effects. Therefore, attentive strategies are needed to prevent or minimize the antagonizing effect of the potential therapeutic target. Identification of ATP release sources, target receptors, and downstream signalling cascade in different pathological settings may help to determine the target.

### 4.1. Targeting ATP Release Channels

Circulating blood contributes to the increase in extracellular ATP and adenosine, which are predominantly released by RBCs in response to ischemia, tissue damage, and inflammation. Ticagrelor (AZD6140), a potent P2Y12 inhibitor, is reported to enhance the release of ATP from RBCs via anion channels, which rapidly degraded to adenosine [208]. Furthermore, ticagrelor prevents adenosine re-uptake by inhibiting the equilibrated nucleoside transporter 1 (ENT1), which contributes to increasing circulating adenosine [209]. These combined effects of ticagrelor collectively increase extracellular adenosine’s inhibitory effects on platelet aggregation via suppression of P2Y12-mediated inhibition of adenylyl cyclase [210] and improve local blood flow in the coronary artery [209]. In acute coronary syndrome (ACS) and ischemic-induced myocardial infarction, which is often associated with platelet aggregation and blood clots, dual anti-platelet therapy (DAPT) consisting of aspirin and P2Y12 receptor antagonist (clopidogrel, prasugrel, ticagrelor) is often prescribed to the patient [211]. Among the P2Y12 antagonists, ticagrelor has superior clinical benefits in potency and less inter-individual variability. A study on the effect of P2Y12 antagonists on VSMCs revealed that oral ticagrelor actively prevented ADP-induced VSMC vasoconstriction compared with clopidogrel and prasugrel treatments [212]. Administration of low-dose aspirin exerts a cardioprotective property by inhibiting platelet cyclooxygenase-1 (COX-1) isoform, subsequently reducing the TxA2-mediated platelet aggregation and vasoconstriction [213]. On the other hand, high-dose aspirin inhibits platelet COX-1 and endothelial COX-2, which are responsible for producing potent vasodilator prostacyclin [214]. Ticagrelor supposedly inhibits the ADP-P2Y12 signalling axis, potentiating prostacyclin-inhibited platelet aggregation [215] (Figure 2). However, in DAPT, a high dose of aspirin reduces the clinical benefits of ticagrelor not only in suppressing platelet aggregation but also attenuating its anti-contractile effect on ADP-induced VCSM contraction [216]. Collective inhibition of prostacyclin with a higher aspirin dose may lead to increased blood pressure and the risk of thrombosis and myocardial infarction.

The hemichannel is the proposed molecular entity responsible for mediating ATP release in both physiological and pathological settings. A report by Kunugi et al. discovered the existence of a negative feedback mechanism of ATP release via maxi-anion channel which strictly regulated ATP-induced ATP release via hemichannels during ischemia [217]. The initial increase in ATP release during the onset of the in vitro ischemic model is via maxi-anion channels. ATP later acted on the P2Y1 receptor to initiate the PLC/IP3/Ca^2+^-dependent NO production. Although NO-mediated connexin channel regulation remains poorly understood, recent studies postulate the possible mechanism involved in this negative feedback. The NO-guanylyl cyclase a1b1 heterodimer (GC1)-cGMP-PKG-PKC signalling cascades potentially delayed hemichannel opening by connexin-43 Ser368 phosphorylation in cardiomyocytes, thereby possibly contributing to reducing the massive release of ATP during ischemic pre-conditioning [218,219,220]. Gap19, a selective connexin-43 hemichannel blocker with no effect on pannexin-1 channels, has been an attractive potential treatment for hypoxic and inflammatory diseases. Recent studies have demonstrated that the protective role of Gap19 counteracts the connexin-43 hemichannel opening by limiting cardiac and cerebral injuries post-I/R [221,222] (Figure 2). 

Additionally, opening pannexin-1 in cardiac myocytes contributes to the accumulation of ATP, which is responsible for the initiation of fibrosis in the peri-infract region of the heart [138]. However, it is important to acknowledge the ATP-induced opening and formation of pannexin-1-P2X7 complex channels also involved in releasing important GPCR- targeted cardioprotectants, such as adenosine and S1P, in ischemic pre- and post-conditioning [147]. Conflicting with this finding, recent work by Yang’s group revealed the involvement of the pannexin-1-ATP-P2Y7 axis in apelin-13-induced cardiomyocyte hypertrophy [223]. The nature of extracellular ATP should be considered from both cellular contexts. In Vessey’s work [147], exogenous ATP (obtained from pharmacological treatment of ATP) induced pannexin1 and P2X7 channel interaction, which required the release of endogenous cardioprotectants. In Yang’s work [197], endogenous ATP released via pannexin-1 caused the opening of P2X7 channels, triggering autophagy and apelin-13-induced cardiomyocyte hypertrophy. Potentially, brief administration of a low concentration of ATP before or after index ischemia could impair the development of cardiomyocyte hypertrophy, which is causally related to myocardial infarction following I/R injury (Figure 2). However, simultaneously targeting the opening and blocking of cardiomyocyte-channel-derived ATP release during pre- and post-ischemia will be challenging therapeutic strategies. 

### 4.2. Targeting ATP-P2X Signalling Axis

Inhibition of P2X7 receptors on VSMCs, ECs, cardiomyocytes, and cardiac fibroblast has been deemed beneficial in attenuating hypertension, excessive inflammatory response, and developing myocardial fibrosis [108,138,187,224]. Potential P2X7 receptor inhibitors developed by AstraZeneca (AZD9056) and Pfizer (CE-224535) have been tested in a clinical trial for rheumatoid arthritis, which potently inhibits ATP-P2X7-induced IL-1β and IL-18 releases in the patient [225,226]. However, both treatments failed to improve symptoms and lower the inflammatory markers in the patients. These clinical findings indicate that targeting the P2X7 receptor alone is inadequate to suppress the deleterious effect of pro-inflammatory cytokines that might be released by other signalling pathways in rheumatoid arthritis. However, targeting P2X7 receptors in cardiovascular diseases may seem promising, because collective activation of P2X7 in myocytes and non-myocytes contributes to an unwarranted inflammatory response that can lead to myocardial infarction. Inhibition of P2X7 receptors in these cells might have a more significant impact on reducing the inflammatory markers. 

In recent years, several animal studies have shown the promising outcome of the P2X7 receptor antagonists. Hansen’s group recently revealed the therapeutic potential of highly specific P2X7 allosteric antagonist PKT100 modulating pulmonary hypertension and right ventricle hypertrophy via suppression of maladaptive ATP/P2X7 axis-derived inflammasome and IL-1β production [227]. Similarly, in cardiac fibroblast, administration of Brilliant Blue G (BBG), another P2X7 antagonist, significantly attenuated TAC-induced cardiac fibrosis via the inhibition of the ATP/P2X7-mediated NLRP3/IL-1β pathway [191]. Additionally, immunosuppressive treatment of the P2X7-specific antagonist A740003 improved myocardial contraction in the murine experimental autoimmune myocarditis model (Figure 2). It is necessary to conduct in-depth studies to understand the safety of these P2X7 antagonists in cardiovascular diseases before further translating into clinical studies.

Another possible target from P2X receptor subtypes is P2X4 receptors. Unlike P2X7, the cardiac P2X4 receptor has been reported by several studies to exert beneficial roles in pathological remodelling. Overexpression of the P2X4 gene and pharmacological treatment of ATP-induced P2X4 in cardiomyocytes and ECs demonstrated a protective role in ischemic and pressure-overload-induced heart failure by tissue-specific activation of eNOS [203,228]. Furthermore, P2X4, one of the most Ca^2+^-permeant channels among the P2X receptors, can increase cardiac contractility and improve cardiac performance, which will be beneficial in combating heart failure progression. Despite this evidence, finding a specific agonist other than ATP for P2X4 receptors remains challenging. Paradoxically, using ATP to evoke the protective role of P2X4 may backfire, as this nucleotide also modulates other P2X and P2Y receptors, and the downstream effectors might contribute to different pathological responses. Overall, it was worth noting that activation of the ATP-P2X4 axis is protective only when the ECs are functioning. However, this role is abrogated once the ECs are damaged, for instance, during atherosclerosis development [116]. Therefore, finding an alternative specific agonist or gene-therapy-mediated overexpression of P2X4 receptors with cell-specific effects might help accentuate the protective role of this receptor. 

In addition to P2X4 and P2X7 receptor subtypes, P2X1 is expressed in coronary smooth muscle cells and cardiomyocytes of a normal and failing heart in humans [14]. Due to the higher sensitivity of ATP compared to other P2X subtypes, the ATP-P2X1 axis is actively involved in vasoconstriction in the vasculature, while in cardiomyocytes, a close association of P2X1 and ATP release channel and connexin-1 was found in patients with dilated cardiomyopathy; the pathological significance remains obscure [229]. Until now, there have been limited studies on ATP-P2X1’s functional role in the pathological setting of cardiovascular remodelling. Several selective agonists and antagonists of human P2X1 receptors have been identified from previous works and recently summarized in the review by Bennetts et al. [230]. Therefore, extensive study needs to be conducted using these agonists and antagonists to better understand the pathophysiological role of these highly ATP-sensitive receptors.

### 4.3. Targeting ATP-P2Y2 and Downstream Effectors

P2Y2 receptors play a central role in modulating various downstream effectors in cardiovascular-related diseases, including hypertension, atherosclerosis, I/R injury, myocardial fibrosis, and atrophy. The previous section highlighted the detrimental activation of the ATP-P2Y2 signalling axis on VSMCs and ECs in hypertension, cardiomyocytes in myocardial atrophy, and cardiac fibroblast in myocardial fibrosis. Despite extensive studies on the development of potent P2Y2 receptor antagonists, most outcomes demonstrated severe drawbacks due to a lack of selectivity, low oral bioavailability, and generally poor metabolic stability and pharmacokinetic properties. However, the introduction of thiouracil derivative AR-C118925 by AstraZeneca around 20 years ago greatly contributed to the pharmacological studies to elucidate the physiological and pathological importance of P2Y subtypes [231]. Due to the vast action of this antagonist on almost all P2Y and all P2X receptors, subsequent works have been conducted by modifying its structure. For instance, the final modifications include adding a symmetric methyl group and monocarboxylic group to increase the activity [231]. This ultimately yielded a potent and selective P2Y2 antagonist where the underlying inhibitory mechanisms are through the suppression of Gq-mediated and Ca^2+^-releasing pathways [232]. In our laboratory, we showed a significant reduction in ATP/P2Y2-induced cardiomyocyte atrophy with the treatment of AR-C118925 [205] (Figure 2). However, further in vitro and in vivo studies are strongly needed to evaluate the potential of this antagonist on the pathological response coupled with the activation of P2Y2 receptors in cardiovascular diseases. 

On the contrary, there is an opposing role of AR-C118925-inhibited P2Y2 in ischemic heart disease. Previous findings showed the protective role of P2Y2 in ischemic heart disease and angiogenesis [143,145,166,233]. A study by Hochhauser’s group demonstrated that the pharmacological inhibition of the P2Y2 receptor by AR-C118925 exacerbates ischemic damage in cardiomyocytes [143]. In contrast, in vitro and in vivo treatment of MRS2768, a moderately potent and selective P2Y2 receptor agonist, attenuates myocyte damage and improves myocardial function in ischemic injury. In another cellular context, MRS2768 shows pro-fibrotic properties on cardiac fibroblast via P2Y2-mediated intracellular Ca^2+^ transient [234]. Hence, in-depth investigation is needed to understand the benefits of activation by MRS2768 or inhibition by AR-C118925 of P2Y2 receptors in the reparative mechanism during ischemic heart disease due to the seemingly cell-specific effect of these compounds.

Given the significant importance of the ATP-P2Y2 axis in pathological and protective roles, targeting the downstream effector may become a suitable option. The Ca^2+^-permeable non-selective cation channel TRPC3 has been identified as an important mediator in ATP-P2Y2-Gq-PLC-dependent Ca^2+^ signalling associated with pathological cardiac remodelling [235,236]. TRPC3 interaction with other membrane-bound proteins has been reported to be associated with muscular dystrophy, myocardial hypertrophy, and atrophy; each was activated with different upstream stimuli [180,186,237]. The severity of TRPC3-Nox2 pathology-specific interaction functions via ATP-P2Y2 signalling, which leads to excessive ROS production, has revealed a promising therapeutic target for myocardial atrophy. A work by Nishiyama and colleagues showed that the interaction of TRPC3-Nox2 in doxorubicin-induced myocardial atrophy is attenuated by the treatment of ibudilast, a phosphodiesterase (PDE) inhibitor [235]. An early study demonstrated the inhibitory effect of ibudilast on PDE and enhanced intercellular cAMP levels to promote a cardioprotective effect [238].

Interestingly, ibudilast also attenuates the co-morbidity effect of doxorubicin treatment in mice with systemic-toxicity-induced muscle atrophy [235]. Recently published work by the same group revealed that ATP released via pannexin-1 in response to SARS-CoV-2 S protein pseudovirus exposure mediates TRPC3-Nox2 formation associated with the increase in ROS and ACE2 gene expression, which are risk factors for myocardial dysfunction [236]. Moreover, the USA initiated a clinical trial of ibudilast to combat acute respiratory distress syndrome in COVID-19 patients [239]. Additionally, other PDE inhibitors have shown potent vasodilatory and anti-hypertensive effects. Rolipram (PDE4 inhibitor), sildenafil (PDE5 inhibitor), and zaprinast (PDE5 and 6 inhibitors) induce relaxation in ET-1-induced vasoconstriction of human mesenteric arteries [240]. Interestingly, celecoxib, a COX-2 inhibitor with unique inhibitory effects on PDE4 and PDE5, also exerts a vasorelaxant effect comparable to other PDE inhibitors in the cAMP- and cGMP-dependent pathways [240] (Figure 2). Collectively, targeting the downstream pathological protein–protein interaction might be more efficient and convenient to minimize the adverse side effects that can be assumed by direct inhibition of the P2Y2 receptor.

## 5. Concluding Remarks

Overall, this review highlights the interconnected mechanism underlying the dual role of ATP in the physiological and pathological remodelling of the heart. In cardiovascular diseases, the involvement of purinergic signalling by ATP is well-connected. ATP released during the development of hypertension and atherosclerosis regulates vascular tone and remodelling. ATP may act as a danger signal by informing other resident cells to initiate the protective mechanism combating the approaching insult. However, accumulating extracellular ATP changes the Ca^2+^ homeostasis in VSMCs and ECs, leading to vascular endothelial damage. More ATP will be released, alerting other resident cells to the incoming changes in the metabolic function of the heart. Low blood flow will decrease the oxygenated blood supply that prompts the I/R injury, predominantly affecting the cardiomyocytes’ viability. Dynamic angiogenesis might be a final resolve to rescue the dying cardiomyocytes by forming new blood vessels to deliver oxygen and essential nutrients. Dying myocytes contribute to high extracellular ATP, which acts on purinergic receptors, thus generating excessive production of inflammatory cytokines and pro-fibrotic factors. The infarcted site undergoes an uncontrolled reparative mechanism by pro-fibrotic factors, leading to myocardial fibrosis. Consequently, the contractility of the cardiomyocytes decreases due to the high ratio of fibroblast to myocytes in the left ventricular, instigating hypertrophic- and atrophic-response-mediated heart failure. 

Due to the pleiotropic role of ATP in these disease settings, it is challenging to target one particular receptor. Understanding the actions and consequences of ATP-modulated P2 receptor subtype activation in a cell-specific perspective may change our perception of designing the therapeutic target. Targeting downstream effectors could be exploited to regulate the detrimental effect of ATP release and accumulation in the extracellular milieu. However, further investigation is required to understand the counter-regulatory activity of ATP on the onset of the change from a protective to a pathological role in cardiac remodelling.

## Figures and Tables

**Figure 1 molecules-28-02102-f001:**
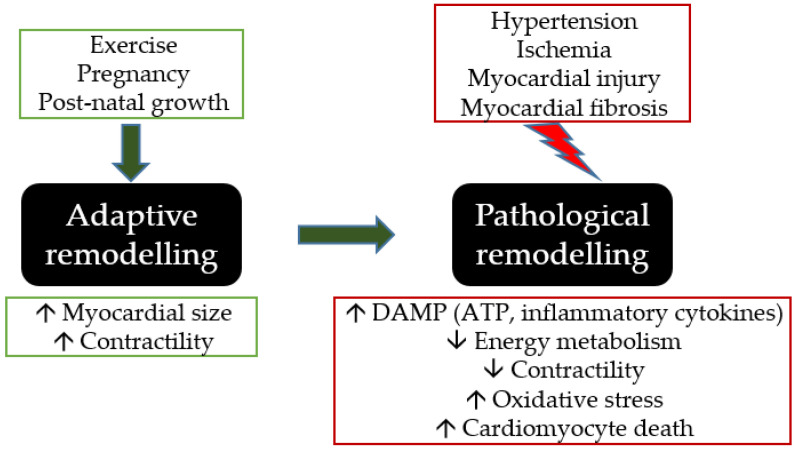
Adaptive and pathological myocardial remodelling.

**Figure 2 molecules-28-02102-f002:**
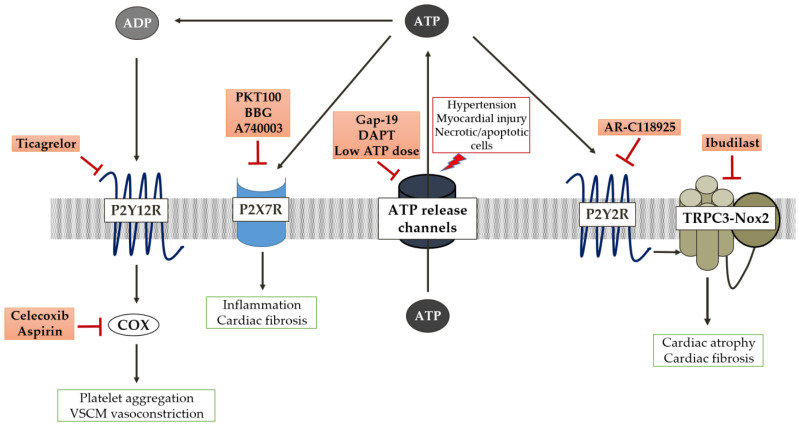
Therapeutic insight of the pleiotropic role of ATP in myocardial remodelling.

**Table 1 molecules-28-02102-t001:** The underlying autocrine and paracrine signalling of extracellular ATP released from myocytes and non-myocytes in the cardiovascular system.

ATP Sources	ATP-Released Channels	Autocrine/Paracrine Signalling of Extracellular ATP
Cardiomyocytes	Connexin-43 Pannexin 1 Pannexin 2 Pannexin-1/P2X7 complex	P2X4 –basal cardiac contractility P2Y/Gq—PLC—↑ Ca^2+^—contraction P2Y11/Gs—cAMP—↑ Ca^2+^—contraction and relaxation
CF	Connexin-43 Connexin-45	P2Y2—α-SMA/TGF-β/PAI-1—sufficient scar formation and inflammatory response
VSCM	Connexin-43 Pannexin-1 ABC transporters	P2X1—basal vascular contractility
EC	Cav-1 VRAC Connexin hemichannels	P2X4—↑ Ca^2+^—NO—vasodilation P2Y/Gq—PLC—↑ Ca^2+^—NO—vasodilation
RBC	CR-1 CTFR Pannexin-1 VDAC	P2Y1—↑ osmolyte permeability—absorption of sufficient nutrient P2X1 and P2X7—PS exposure—hemolysis

**Table 2 molecules-28-02102-t002:** Involvement of ATP-purinergic signalling in protective and pathological response of cardiovascular remodelling.

Pathological Condition	Protective Signalling	Pathological Signalling
Hypertension and Atherosclerosis	ATP-P2X4-NO-Vasodilation ATP→Adenosine-A2A-NO-Vasodilation	ATP-P2X7-NLRP3-Pro-inflammatory cytokines ATP-P2X1-Vasocontriction ATP-P2X1/P2X4-Vasocontriction ATP-P2Y2-VSMC migration/proliferation/hypertrophic vascular remodelling
Ischemic/ Reperfusion Injury	ATP/P2Y2-AKT, ERK-Cardiomyocyte survival ATP-Pannexin1/P2X7-Cardioprotectant (SP1, adenosine) ATP→Adenosine-A2B-HIF1α-protection against hypoxia	ATP-P2X7-NLRP3-Pro-inflammatory cytokines/fibroblast differentiation ATP-P2Y6-Vascular inflammation
Angiogenesis	ATP/P2Y2-AKT, ERK, mTOR-VEGF-Angiogenesis ATP/VEGF_165_-Angiogenesis ATP/P2Y2-miR-22-ICAM-1 downregulation-tumourgenesis suppression	ATP/P2Y2-CTGF, VEGF, VCAM-1-TNBC progression and metastasis ATP/P2X7-VEGF-Angiogenesis
Myocardial fibrosis	ATP/P2Y1-reduce ERK-Impaired production of pro-fibrotic factors	ATP-P2X7-NLRP3-Pro-inflammatory cytokines/fibroblast differentiation ATP-P2Y2-TRPC3-replacement fibrosis ATP-P2Y2-MAPK-pro-fibrotic factors
Myocardial hypertrophy and atrophy	ATP/P2Y2-TRPC5/eNOS-inhibit cardiomyocyte hypertrophy ATP-P2X4-eNOS-inhibit cardiomyocyte hypertrophy	ATP-P2Y2-TRPC3/Nox2-cardiomyocyte atrophy

## Data Availability

Not applicable.

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
