# Peer review of "The Pleiotropic Role of Extracellular ATP in Myocardial Remodelling"

_molecules, 2023, doi:10.3390/molecules28052102_

Round 1

Reviewer 1 Report

Left ventricular remodeling is an adaptive process modifying the ventricular wall thickness, ventricular volume, shape, mass and structure of the myocardium. In 1990, Pfeffer and Braunwald published a review on cardiac remodeling following myocardial infarction. Over the past decade, clinical and basic research on ventricular remodeling has significantly progressed.

To begin with, upon first reading the manuscript, I noticed a great deal of work done by the team for data collection and for the elaboration of this manuscript. The aim of this review article was to investigate a better understanding of ATP communication in myocardial remodelling

The presented data are of high quality and convincing, as well as very relevant. The manuscript is very well-written and properly documented.

In my opinion the current version of the manuscript should be enriched in order to increase the interest of the reader. I would suggest to adding information (in the form of two graphic abstracts) in the introduction and in the endof this manuscript aboutsummary

-          of role of ATP in cardiovascular remodelling (in the introduction)

-          of therapeutic insights of ATP signalling in cardiac remodelling (in the end)

Moreover, please add the information from research articles (The presented information will be more persuasive as well as more relevant for the development of new therapeutic methods), what describe:

-          comparison effects of various doses of aspirin coadministered with the highly effective P2Y12-inhibitor on the reactivity of vascularsmooth muscle cells

-          comparison the vasodilating properties of phosphodiesteraseinhibitors, cyclooxygenase‑2 inhibitors or other drugs in arteries constricted with endothelin‑1

-          comparison effects of different P2Y12 antagonists on the reactivityof vascular smooth muscle cells

Reviewer 2 Report

This is an excellent review of the pleiotropic role of extracellular ATP in myocardium remodelling. It is a novel review highlighting the importance of understanding the underlying roles of extracellular ATP in pathological remodelling to find new therapeutic targets and combat heart failure.

·         Writing is straightforward and comprehensive. Furthermore, it fills a gap in knowledge, as many articles in the field have been published over the years, but the most important findings are summarized here.

·         Consider adding some figures to summarise and connect all the information given in the review.

Reviewer 3 Report

The authors of the manuscript ‘The Pleiotropic Role of Extracellular ATP in Myocardial Remodelling’describe the current understanding of the vital role of ATP and purinergic receptors in the physiology of heart, and especially in its remodeling. The subject is interesting and falls well within the scope of the Journal.

As I mentioned, in principle, the manuscript is potentially interesting but I have some suggestions:

·      Cardiomyocytes rely on an efficient blood supply delivered by the vasculature. Hence, increasing angiogenesis is a vital element in e.g. heart failure. Also, three main phases widely recognized following myocardiac infarction are inflammation, angiogenesis and fibrosis. Meanwhile, the authors did not touch at all angiogenesis, which is crucial in this topic. Therefore, I suggest that the authors add a chapter on the participation of ATP and GPCRs in the process of angiogenesis. There is a lot of literature on this subject for example doi: 10.1074/jbc.M401799200, doi: 10.1038/sj.bjc.6604998, doi: 10.1016/j.bbrc.2011.12.088,  doi: 10.3390/ijms23010238 

·      In the current form, the text is difficult to trace. It’s quality would be markedly improved by the authors after preparation of several Figures  and a Tables. 

·      Typos, e.g. 

G-protein-coupled instead of G protein-coupled purinoceptors

2-mesATP instead of 2-MeSATP

·      The same abbreviation is expanded several times in the text, e.g.

These receptors are G-protein coupled 140 receptors (GPCRs) that form a specific isoform with either Gq, Gs, and G12/13 alpha sub-141 unit [38,39]. The coupling of these receptors with Gq protein initiates the phospholipase

These receptors are G-protein coupled 140 receptors (GPCRs) that form a specific isoform with either Gq, Gs, and G12/13 alpha sub-141 unit [38,39]. The coupling of these receptors with Gq protein initiates the phospholipase

logical and pathological processes in the cardiovascular system. Adenosine triphosphate 50 (ATP) is an organic chemical that regulates numerous physiological mechanisms by acting 51 on purinergic receptors [4,5]. In 1972, purinergic signalling was first proposed, where 52 adenosine triphosphate (ATP)

Round 2

Reviewer 3 Report

The authors improved the content as requested.